# Comparison of Characteristics of Definition Criteria for Respiratory Sarcopenia—The Otassya Study

**DOI:** 10.3390/ijerph19148542

**Published:** 2022-07-13

**Authors:** Takeshi Kera, Hisashi Kawai, Manami Ejiri, Kumiko Ito, Hirohiko Hirano, Yoshinori Fujiwara, Kazushige Ihara, Shuichi Obuchi

**Affiliations:** 1Department of Physical Therapy, Takasaki University of Health and Welfare, Gunma 370-0033, Japan; 2Research Team for Human Care, Tokyo Metropolitan Institute of Gerontology, Tokyo 173-0015, Japan; hkawai@tmig.or.jp (H.K.); ejiri@tmig.or.jp (M.E.); kumiko@tmig.or.jp (K.I.); obuchipc@tmig.or.jp (S.O.); 3Research Team for Promoting Independence and Mental Health, Tokyo Metropolitan Institute of Gerontology, Tokyo 173-0015, Japan; h-hiro@gd5.so-net.ne.jp; 4Research Team for Social Participation and Community Health, Tokyo Metropolitan Institute of Gerontology, Tokyo 173-0015, Japan; fujiwayo@tmig.or.jp; 5Department of Social Medicine, Hirosaki University School of Medicine, Aomori 036-8562, Japan; ihara@hirosaki-u.ac.jp

**Keywords:** forced vital capacity, peak expiratory flow rate, respiratory muscle strength, respiratory sarcopenia, sarcopenia

## Abstract

We compared the definitions of respiratory sarcopenia obtained from a model based on forced vital capacity (FVC) and whole-body sarcopenia, as recommended by the Japanese Association of Rehabilitation Nutrition (JARN), and a model based on the peak expiratory flow rate (PEFR), as recommended in our previous study. A total of 554 community-dwelling older people without airway obstruction who participated in our study in 2017 were included in the current study. Respiratory function, sarcopenia, and frailty were assessed. Pearson’s correlation coefficients of the associations of the FVC and PEFR with physical performance and the receiver operating curves of FVC and PEFR’s association with sarcopenia, long-term care insurance (LTCI) certification, and frailty were calculated. The sensitivity and specificity of the two respiratory sarcopenia models were assessed. The FVC and PEFR were associated significantly with physical performance. The areas under the curve for sarcopenia and the LTCI certification in the FVC and PEFR groups were statistically significant in both sexes. While Kera’s model had a lower specificity in determining sarcopenia, it had a sensitivity higher than the JARN model. Both models provide suitable definitions of respiratory sarcopenia. Future studies are required to determine other appropriate variables to define respiratory sarcopenia.

## 1. Introduction

Sarcopenia is an important conditions that becomes more apparent with aging. It is defined by a decline in skeletal muscle mass and function [1,2]. The decline in respiratory muscle function that is observed with aging is similar to that observed in the other skeletal muscles [3,4,5]. Thus, respiratory sarcopenia should be defined as the decrease in respiratory muscle mass and function. However, this definition of respiratory muscle sarcopenia is burdensome since it is difficult to measure the loss of respiratory muscle mass.

Respiratory muscle strength is usually evaluated using the maximal inspiratory or expiratory mouth pressure (MIP or MEP). The MIP and MEP are based on maximal voluntary inspiration and expiration (Mueller and Valsalva maneuvers) [6,7], respectively. Furthermore, the use of special equipment, such as a mouth pressure meter or pressure transducer, is required for measuring mouth pressure [8]. While the respiratory muscle mass can be measured using magnetic resonance imaging and computed tomography [9], the requirement of complex equipment makes the measurement process difficult, especially in community and clinical settings.

The peak expiratory flow rate (PEFR), which is related to body composition and physical function [10], decreases with age [11], as does the appendicular skeletal muscle mass (ASM). Our previous study showed a significant relationship between the PEFR and both sarcopenia and long-term care insurance (LTCI) certification and recommended that the definition of respiratory sarcopenia be based on PEFR [12].

Recently, the Japanese Working Group on Respiratory Sarcopenia of the Japanese Association of Rehabilitation Nutrition (JARN) published the diagnostic criteria for respiratory sarcopenia and acknowledged it as an aging-related functional disability [13]. They defined respiratory sarcopenia as a “whole-body sarcopenia and low respiratory muscle mass followed by low respiratory muscle strength and/or deteriorated respiratory function.” Specifically, with regard to low respiratory muscle strength, they defined respiratory sarcopenia as a low MIP/MEP or low forced vital capacity (FVC) and low skeletal muscle mass, muscle strength, and function using the cut-offs specified by a previous consensus statement on sarcopenia. However, the PEFR has not been accepted widely as a diagnostic criterion for respiratory sarcopenia, due to the possible effect of airway obstructions [13]. Nevertheless, the PEFR has some advantages over the FVC, as it is not influenced by the mechanical properties of the respiratory system, including the lungs, muscles, joints, and ligaments of the thoracic cage. This is the first study to compare two respiratory sarcopenia models.

This study aimed to compare and analyze the definitions of respiratory sarcopenia obtained using the JARN’s recommended FVC-, skeletal muscle mass-, and function-based model and the PEFR-based model recommended in previous studies. Moreover, we aimed to show that those definitions are plausible definitions of respiratory sarcopenia.

## 2. Materials and Methods

### 2.1. Participants

The Tokyo Metropolitan Institute of Gerontology initiated a comprehensive health checkup program called “The Otassha Study” at the Itabashi Ward of Japan in 2011 [12]. Overall, 761 (319 men and 442 women) community-dwelling older adults participated in the program in 2017 (Figure 1). The participants who completed the assessment for respiratory function, sarcopenia, frailty, walk speed, grip strength, and body composition and who underwent spirometry were included in this study; patients with chronic obstructive pulmonary disorder and airway obstruction were excluded.

### 2.2. Measurement of Physical and Respiratory Function and Questionnaire

In Japan, the long-term care insurance (LTCI) system was designed to classify older people according to their conditions that required support or long-term care. There are seven levels of long-term care: support levels 1 and 2 and care need levels 1 (least disabled) to 5 (most disabled). Thus, the LTCI category indicated the level of user’s physical or cognitive function. The participants were asked to specify whether they categorized by the LTCI as requiring support or not [14]. The participants were also requested to complete a questionnaire based on the Kihon checklist (KCL) [15]; the KCL [16] consists of 25 items and is used to assess frailty.

A manual stopwatch was used to determine the gait speed along a 5 m course with a 3 m acceleration and deceleration area. The grip strength while standing was measured using a Smedley-type grip strength meter. The body composition was measured using a bioelectrical impedance analyzer (Model InBody 720, InBody, Seoul, Korea). The ASM was calculated as the sum of the lean soft tissue masses of the arms and legs.

The respiratory function was measured using spirometry with an electronic spirometer (Autospiro AS-507, Minato Medical Science, Osaka, Japan). The FVC, PEFR, and forced expiratory volume in one second (FEV_1_) were measured. The FEV_1_/FVC was also computed to determine the possible influence of airway obstruction [17].

### 2.3. Definition of Sarcopenia and Frailty

Sarcopenia was defined according to the criteria outlined by the Asian Working Group for Sarcopenia 2019 (AWGS) [18]. The cut-off values for sarcopenia were as follows: ASM/height^2^ of <7.0 kg/m^2^ for men and <5.7 kg/m^2^ for women, grip strength of <28 kg for men and <18 kg for women, and gait speed of <1.0 m/s for both sexes. We did not differentiate between possible sarcopenia, sarcopenia, and severe sarcopenia; therefore, participants with sarcopenia of the whole body and low grip strength or low gait speed were considered to have AWGS sarcopenia in this study. Frailty was determined using the total KCL score and Satake’s criteria (cut-off point, 7/8), whereby participants who met eight or more criteria were considered as frail [19].

### 2.4. Definition of Respiratory Sarcopenia

Two respiratory sarcopenia models were constructed: (a) low FVC + sarcopenia of whole-body (low ASM/height^2^ + low walk speed and/or low grip strength) (the JARN model) [13] and (b) low PEFR (Kera’s model) [12]. In the flowchart for used to diagnose respiratory sarcopenia according to the JARN, respiratory sarcopenia is defined using skeletal muscle mass, maximal mouth pressure [13], FVC, and respiratory muscle mass. However, since it is difficult to measure respiratory muscle mass, we defined both “definite” and “probable” respiratory sarcopenia for the JARN model, with the lower limit of the normal FVC being set as the cut-off based on the Lambda Mu Sigma method, which allows for the simultaneous modeling of the mean (*μ*), coefficient of variation (*σ*), and skewness (*λ*) of a distribution [20]. The PEFR cut-off was set at <4.40 L/s for men and <3.21 L/s for women [12].

### 2.5. Statistical Analysis

To determine the relationship between the candidate variables used for defining respiratory sarcopenia and some “aspects of frailty” related to aging and to determine the association between the respiratory function and the variables used in the AWGS definition of sarcopenia, Pearson’s correlation coefficients were calculated. The receiver operating characteristic (ROC) curves of the FVC and PEFR for AWGS sarcopenia, LTCI, and frailty were calculated for each sex. The overall sensitivity and specificity of each respiratory sarcopenia for AWGS sarcopenia, LTCI, and frailty were calculated for both of the JARN and Kera models using the cut-off points for each sex. The continuous variables were presented as means  ±  standard deviations. The numerical data were described as percentages (%). SPSS version 26.0 (IBM, Armonk, NY, USA) was used to perform the statistical analyses.

## 3. Results

In total, 554 participants (226 men; age 72.8 ± 6.5 years, 328 women; age 73.5 ± 6.1 years) who met the study criteria were included in the analysis. Of these, 21 participants (3.8%) were diagnosed with respiratory sarcopenia using the JARN model, and 91 participants (16.4%) using Kera’s model. The characteristics of respiratory sarcopenia in men and women are described in Table 1.

The Pearson’s correlation coefficients for the associations between respiratory function, physical performance, and the KCL score for each sex are shown in Table 2. The FVC and PEFR were related significantly to physical performance and the KCL score; however, the FEV_1_/FVC, as an indicator of airway obstruction, was not related to these variables. The FVC was strongly related to vital capacity (VC), and the PEFR was slightly related to the FEV_1_/FVC in both sexes.

The ROC analysis results for both sexes are shown in Table 3 and Figure 2. The areas under the curve (AUCs) of the FVC and PEFR for AWGS sarcopenia and “certified for LTCI” showed a significant relationship in both sexes (AWGS sarcopenia, AUC: 0.67–0.76, *p* = 0.002– <0.001; LTCI, AUC: 0.70–0.90, *p* = 0.023– <0.001). The AUCs of the FVC and PEFR for frailty were lower than those for AWGS sarcopenia and LTCI and were not significant except for the FVC in men. Kera’s model had lower specificity for AWGS sarcopenia, LTCI, and frailty but had higher sensitivity than the JARN model, as shown in Table 4.

## 4. Discussion

The association between the FVC and PEFR was strong with regard to physical performance and frailty status. Although the JARN model did not provide direct evidence for the proposed concept and diagnostic criteria for respiratory sarcopenia, we showed that the FVC was associated with sarcopenia, and that the assessment of the respiratory function using spirometry is particularly important in the definition of respiratory sarcopenia.

We did not expect the FVC to have a higher AUC higher than the PEFR with respect to the external criteria because the FVC may have been influenced by the mechanical properties of the respiratory system. The FVC relies on compliance of the respiratory system compliance and respiratory muscle strength, similar to the VC, except when the airway is obstructed. Furthermore, the FVC has a positive relationship with increasing respiratory system compliance and respiratory muscle strength; the higher the compliance and respiratory muscle strength, the greater the increase in the FVC. Both factors are known to deteriorate with aging [14]. However, the FVC is affected more by the compliance of the respiratory system than by respiratory muscle strength when considering the relationship between lung volume and expiratory pressure, as denoted by the “S” curve that has been shown with spirometry [21]. The FVC, which is defined by the differential lung volume at the maximal inspiratory and expiratory positions, requires a high compliance of the respiratory system and high respiratory muscle strength, for the expansion and/or compression of the rib cage; however, the compliance of the respiratory system reduces with aging. Therefore, the FVC may decrease when the chest wall, muscles, and joints become stiff with aging (reducing the compliance of the respiratory system), even if the respiratory muscle strength remains high. In contrast, the PEFR depends on the individual’s effort and respiratory muscle strength. The PEFR observed at initial expiration is only affected minimally by the compliance of the respiratory system.

The JARN model indicated that the PEFR was not acceptable as a diagnostic criterion for respiratory sarcopenia it has an air-flow limitation [13]. In fact, it has been difficult to determine sarcopenia using the PEFR in patients with chronic obstructive pulmonary disease, in which the PEFR relies on an airway obstruction rather than a respiratory weakness [22]. The weak but significant relationship between the PEFR and the FEV_1_/FVC observed in our study raises concerns. Given that the JARN concept of respiratory muscle sarcopenia, in addition to the original model of sarcopenia, is based on low FVC a strong association between FVC and sarcopenia in the ROC analysis was expected; however, the sensitivity of Kera’s model in identifying sarcopenia, frailty, and certification of LTCI was higher than that of the JARN model, although it did not include variables related to skeletal muscle mass. It also indicates the superiority of the respiratory sarcopenia model based on the PEFR.

According to a previous study on the relationship between sarcopenia and respiratory muscle strength, there was low respiratory muscle strength in sarcopenia [23]. Furthermore, when sarcopenia was set as an independent variable, the AUCs of MIP (men: 0.793, women: 0.734) and MEP (men: 0.739, women: 0.715) were moderate, indicating a significant relationship between sarcopenia and respiratory muscle strength [22]. On the other hand, in the current study, the AUCs of the PEFR and FVC for sarcopenia were similar to those of the MIP and MEP, providing indirect evidence for the consideration of the PEFR and FVC as alternatives to the MIP and MEP. Hence, both the FVC and PEFR are suitable for use as operational definitions of respiratory sarcopenia. However, the FVC was directly related to the VC [24], and the correlation coefficient was above 0.9. Considering the effort needed for expiration when measuring the FVC and that the FVC and VC showed almost the same value under the condition of absence of airway obstruction, we believe it is better to use the VC, which is less burdensome to measure, instead of the FVC in the JARN model. In general, the PEFR can be measured not only using spirometry, but also using the plastic peak flow meter used for asthma management. The PEFR is easier to measure than the FVC. Since there is a predominance of PEFR in the clinical setting, it is recommended that the PEFR be used as the definition of respiratory sarcopenia.

This study had some limitations. As “declining skeletal muscle mass” was not included in Kera’s model, it may not be suitable in terms of the original sarcopenia concept [1]. Further analysis is required to identify the specific parameters associated with the diagnosis of respiratory sarcopenia.

## 5. Conclusions

Although the results of our study may not have been sufficient to dispel the concerns that arose out of the use of the JARN model and the effect of the airway obstruction that was observed in our respiratory model using the PEFR, both the JARN and Kera models may be suitable to define respiratory sarcopenia. We highlight the importance of defining respiratory sarcopenia, as it can predispose patients to complex diseases, aspiration, and death. We believe that this preliminary study provided a basis for further research on the definition of respiratory sarcopenia.

## Figures and Tables

**Figure 1 ijerph-19-08542-f001:**
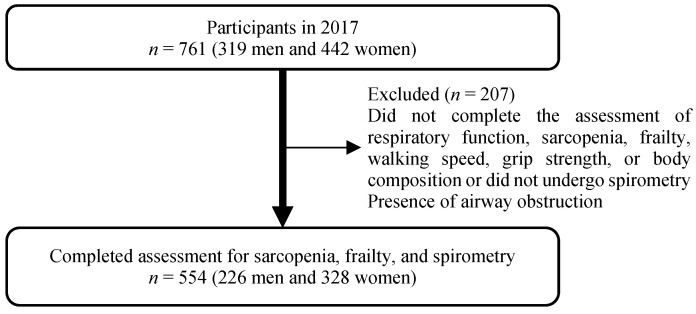
Flowchart of the study participants.

**Figure 2 ijerph-19-08542-f002:**
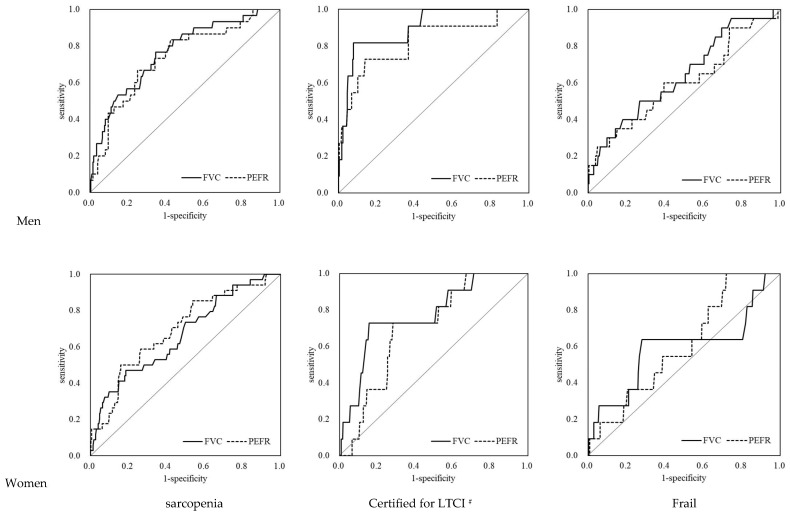
ROC analysis of FVC and PEFR for criterion variables. ^#^ Support levels 1 or 2 and long-term care levels 1–5 on LTCI. FVC: forced vital capacity; PEFR: peak expiratory flow rate; LTCI: long-term care insurance certification.

**Table 1 ijerph-19-08542-t001:** Characteristics of respiratory sarcopenia between JARN and Kera models by sex.

**Low FVC and sarcopenia (JARN model)**						
	**Men**			**Women**		
	**Robust** **(*n* = 216)**	**Respiratory** **sarcopenia** **(*n* = 10)**	***p*-value**	**Robust** **(*n* = 317)**	**Respiratory** **sarcopenia** **(*n* = 11)**	***p*-value**
Age, year	72.5 (6.4)	78.8 (6.1)	0.005	73.3 (6.0)	80.6 (6.0)	0.001
BMI, kg/m^2^	23.7 (3.3)	20.7 (2.1)	0.003	22.4 (3.2)	21.4 (2.3)	0.300
Grip strength, kg	34.6 (7.1)	24.7 (5.2)	<0.001	22.4 (4.6)	14.8 (2.0)	<0.001
Walk speed (usual), m/s	1.43 (0.27)	1.17 (0.25)	0.005	1.44 (0.24)	1.21 (0.2)	0.002
ASM/ht^2^, kg/m^2^	7.51 (0.78)	6.62 (0.24)	<0.001	5.94 (0.71)	5.37 (0.18)	<0.001
PEFR, L/s	6.54 (1.81)	4.29 (1.51)	<0.001	4.45 (1.11)	2.98 (1.44)	0.001
FVC.L	3.04 (0.57)	2.06 (0.33)	<0.001	2.1 (0.37)	1.48 (0.18)	<0.001
VC, L	3.24 (0.64)	2.18 (0.41)	<0.001	2.22 (0.41)	1.60 (0.19)	<0.001
%VC, %	92.3 (14.7)	68.2 (7.4)	<0.001	94.2 (14.2)	74.9 (5.5)	<0.001
FEV_1_, L	2.42 (0.46)	1.63 (0.3)	<0.001	1.69 (0.31)	1.14 (0.14)	<0.001
%FEV_1_, %	88.6 (14.1)	67.9 (12.4)	<0.001	98.6 (14.6)	76.7 (9.7)	<0.001
**Low PEFR (Kera’s model)**						
	**Men**			**Women**		
	**Robust** **(*n* = 192)**	**Respiratory** **sarcopenia** **(*n* = 34)**	***p*-value**	**Robust** **(*n* = 271)**	**Respiratory** **sarcopenia** **(*n* = 57)**	***p*-value**
Age, year	71.9 (6.1)	77.7 (6.9)	<0.001	73 (5.8)	76.2 (6.8)	<0.001
BMI, kg/m^2^	23.6 (3.2)	23.1 (3.9)	0.408	22.4 (3.2)	22.3 (3.2)	0.835
Grip strength, kg	35.1 (6.9)	29.4 (7.5)	<0.001	22.7 (4.6)	19.6 (4.7)	<0.001
Walk speed (usual), m/s	1.44 (0.27)	1.27 (0.28)	0.001	1.45 (0.24)	1.33 (0.26)	<0.001
ASM/ht^2^, kg/m^2^	7.52 (0.77)	7.20 (0.83)	0.031	5.98 (0.67)	5.64 (0.81)	0.001
PEFR, L/s	6.97 (1.46)	3.47 (0.63)	<0.001	4.77 (0.87)	2.66 (0.54)	<0.001
FVC, L	3.08 (0.56)	2.53 (0.58)	<0.001	2.14 (0.36)	1.76 (0.31)	<0.001
VC, L	3.30 (0.62)	2.61 (0.61)	<0.001	2.28 (0.4)	1.86 (0.34)	<0.001
%VC, %	93.2 (14.1)	80.6 (17.3)	<0.001	95.7 (14.1)	83.5 (11.8)	<0.001
FEV_1_, L	2.46 (0.45)	1.93 (0.44)	<0.001	1.73 (0.3)	1.41 (0.26)	<0.001
%FEV_1_, %	89.3 (13.6)	78.6 (16.8)	<0.001	99.8 (14.4)	88.5 (14.1)	<0.001

PEFR: peak expiratory flow rate; FVC: forced vital capacity; FEV_1_: forced expiratory volume in one second; ASM: appendicular skeletal muscle mass; PEFR: peak expiratory flow rate. Data are expressed as mean (SD).

**Table 2 ijerph-19-08542-t002:** Pearson’s correlation coefficient between respiratory function, physical performance, and Kihon checklist points.

**Men**	**FVC**	**VC**	**%VC**	**PEFR**	**FEV_1_/** **FVC**	**ASM/** **height^2^**	**Grip strength**	**Walking speed**	**KCL**
FVC	1.000	0.939 **	0.772 ***	0.494 ***	−0.056	0.317 ***	0.517 ***	0.309 ***	−0.201 **
VC		1.000	0.847 ***	0.509 ***	−0.037 **	0.358 ***	0.553 ***	0.379 ***	−0.219 **
%VC			1.000	0.393 ***	−0.016	0.151 *	0.344 ***	0.298 ***	−0.226 **
PEFR				1.000	0.206 **	0.185 **	0.380 **	0.308 **	−0.190 **
FEV_1_/FVC					1.000	−0.015	0.032	0.067	−0.055
ASM/height^2^						1.000	0.446 ***	0.100	−0.089
Grip strength							1.000	0.328 ***	−0.273 ***
Walking speed								1.000	−0.346 ***
KCL									1.000
**Women**	**FVC**	**VC**	**%VC**	**PEFR**	**FEV_1_/** **FVC**	**ASM/** **height^2^**	**Grip strength**	**Walking speed**	**KCL**
FVC	1.00	0.931 ***	0.743 ***	0.589 ***	−0.052	0.164 **	0.440 ***	0.270 ***	−0.045
VC		1.000	0.844 ***	0.589 ***	−0.054	0.225 ***	0.438 ***	0.285 ***	−0.046
%VC			1.000	0.434 ***	−0.094	0.059	0.225 ***	0.143 **	0.040
PEFR				1.000	0.168 **	0.234 ***	0.349 ***	0.224 ***	−0.053
FEV_1_/FVC					1.000	0.012	−0.019	−0.071	0.050
ASM/height^2^						1.000	0.392 ***	0.152 **	−0.116 *
Grip strength							1.000	0.350 ***	−0.239 ***
Walking speed								1.000	−0.361 ***
KCL									1.000

ASM: appendicular skeletal muscle mass; FVC: forced vital capacity; FEV_1_: forced expiratory volume in one second; KCL: Kihon checklist score; PEFR: peak expiratory flow rate. * *p* < 0.05; ** *p* < 0.01; *** *p* < 0.001.

**Table 3 ijerph-19-08542-t003:** ROC analysis of FVC and PEFR for criterion variables.

		Men		Women	
		AUC	*p*	AUC	*p*
AWGS sarcopenia	FVC	0.76 (0.67–0.85)	<0.001	0.67 (0.57–0.76)	0.002
	PEFR	0.74 (0.64–0.83)	<0.001	0.69 (0.60–0.79)	<0.001
Certified for LTCI ^#^	FVC	0.90 (0.81–0.98)	<0.001	0.77 (0.63–0.91)	0.002
	PEFR	0.82 (0.68–0.97)	<0.001	0.70 (0.58–0.82)	0.023
Frail	FVC	0.64 (0.51–0.77)	0.040	0.59 (0.38–0.80)	0.310
	PEFR	0.60 (0.46–0.74)	0.132	0.60 (0.46–0.75)	0.242

^#^ Support levels 1 or 2 and long-term care levels 1–5 on LTCI. FVC: forced vital capacity; PEFR: peak expiratory flow rate; LTCI: long-term care insurance certification; AWGS: Asian Working Group for Sarcopenia 2019; ROC: receiver operating characteristics; AUC: area under the curve.

**Table 4 ijerph-19-08542-t004:** Sensitivity and specificity of each type of respiratory sarcopenia for AWGS sarcopenia, LTCI, and frailty.

	Low FVC and Sarcopenia of Whole-Body (JARN Model)			Low PEFR (Kera′s Model)		
	AWGS Sarcopenia	LTCI	Frail	AWGS Sarcopenia	LTCI	Frail
Sensitivity, %	32.8 (21.6–45.7)	27.3 (10.7–50.2)	16.1 (5.5–33.7)	42.2 (29.9–55.2)	50.0 (28.2–71.8)	25.8 (11.9–44.6)
Specificity, %	100.0 (99.3–100.0)	97.2 (95.4–98.4)	96.9 (95.1–98.2)	86.9 (83.6–89.8)	85.0 (81.6–87.9)	84.1 (80.7–87.2)
Positive likelihood ratio	Infinity	9.67 (4.15–22.52)	4.70 (1.87–11.83)	3.22 (2.23–4.65)	3.33 (2.09–5.29)	1.63 (0.87–3.05)
Negative likelihood ratio	0.67 (0.57–0.80)	0.75 (0.58–0.97)	0.87 (0.70–1.00)	0.66 (0.50–0.80)	0.59 (0.40–0.90)	0.88 (0.71–1.09)

FVC: forced vital capacity; ASM: appendicular skeletal muscle mass; JARN: Japanese Association of Rehabilitation Nutrition; LTCI: long-term care insurance certification; PEFR: peak expiratory flow rate. Data are expressed as mean (95% confidence interval).

## Data Availability

The datasets generated and analyzed during the current study are not publicly available due to patient privacy and confidentiality. They are not available on request from the corresponding author.

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
