# Peer review of "Comparison of Characteristics of Definition Criteria for Respiratory Sarcopenia—The Otassya Study"

_ijerph, 2022, doi:10.3390/ijerph19148542_

Round 1

Reviewer 1 Report

The manuscript “Comparison of Characteristics of Definition Criteria for Respiratory Sarcopenia” by Takeshi Kera et.al., summarizes a study to compare the definitions for respiratory sarcopenia. The concept and idea of the work is good, but I would request the authors to revisit their manuscript and readdress all their data collected as tables and make visually appealing and easy to understand graphs and  figures. A lot has been presented in text, but the main idea of it is missing as readers do not have time to spend hours in a manuscript. And in such cases, the manuscript loses its scientific point, despite of the work being good like yours. I will proceed with a decision once changes are addressed. 

1         Kindly address these few points before discussion of each section? Each of the below listed questions should be the way to address any result or discussion. This way the readers can understand the propose or motto of doing that review rather than presenting your flowcharts.

a.       What is the purpose or goal of this manuscript, section, sub-section?

b.       What new information does the result give?

c.       What methods were used to evaluate?

d.       What references were used to support the results?

Kindly re-address each section with the above questions. 

2         References. Kindly elaborate more on the sub-sections with references. Appropriate references are required to support the claim and results of any review.

Author Response

We wish to express our strong appreciation to the reviewers for their insightful comments on our paper. We feel the comments have helped us significantly improve the paper.

Reviewer 1

Comments and Suggestions for Authors

The manuscript “Comparison of Characteristics of Definition Criteria for Respiratory Sarcopenia” by Takeshi Kera et.al., summarizes a study to compare the definitions for respiratory sarcopenia. The concept and idea of the work is good, but I would request the authors to revisit their manuscript and readdress all their data collected as tables and make visually appealing and easy to understand graphs and figures. A lot has been presented in text, but the main idea of it is missing as readers do not have time to spend hours in a manuscript. And in such cases, the manuscript loses its scientific point, despite of the work being good like yours. I will proceed with a decision once changes are addressed.

Response: Thank you for your positive comments. As you indicated, we could not visually show our positive results. To address this, the flow chart of the study participants and ROC curves are shown in the Figure. Our responses to other concerns are provided below.

1 Kindly address these few points before discussion of each section? Each of the below listed questions should be the way to address any result or discussion. This way the readers can understand the propose or motto of doing that review rather than presenting your flowcharts.

  1. What is the purpose or goal of this manuscript, section, sub-section?

Response: Thank you for your comment. We have revised the purpose of the study (lines 66-70).

  1. What new information does the result give?

Response: JARN did not recommend respiratory muscle sarcopenia with PEFR due to airway obstruction. However, the non-dominance of PEFR was unclear when the two methods were compared. In contrast, FVC is almost the same variable as VC; therefore, considering the measurement effort, it is the same as when VC is used. In addition, the definition of JARN requires measurements for the diagnosis of sarcopenia. The novelty of this study is that the definition proposed using PEFR has the advantage of being clinically easy to diagnose, even though it is comparable to FVC in diagnosis. We have added this information before the limitations of the study (lines 228-234).

  1. What methods were used to evaluate?

Response: We did not completely understand the reviewer's intent. Respiratory muscle sarcopenia is a concept for which there is no clear definition yet. JARN has proposed a definition, but it has not been validated. Previous studies referred to the respiratory sarcopenia assessed by PEFR as respiratory muscle “weaknesses” and “other weaknesses,” such as sarcopenia and frailty, and justified them. The present study evaluated and compared the JARN’s and Kera’s models in the same way. Regarding the calculation of sensitivity and specificity, which was one of the evaluation methods, the method was unclear; therefore, the statistical analysis section has been revised (lines 130-132).

  1. What references were used to support the results?

Response: Thank you for your comment. We have added references to support our results.

Kindly re-address each section with the above questions.

2 References. Kindly elaborate more on the sub-sections with references. Appropriate references are required to support the claim and results of any review.

Response: Thank you for your comment. We have added some references.

Reviewer 2 Report

Thank you for inviting me to review this manuscript on “Comparison of Characteristics of Definition Criteria for Respiratory Sarcopenia.” I think the topic is very important for readers in the field of geriatric. However, there are several significant concerns in the study.

Line 24

Is “frailty” correct? In Table 3, female frailty is not significant.

Line 26

There is no description of what “higher than” is about.

Line 59

Although MIP/MEP is mentioned, MEP is not included in the method of evaluation of respiratory sarcopenia according to reference number 13.

Line115-117

It would be easier for readers to understand if the specific formula for the cutoff value of FVC is described. Please consider it.

Table 3

Is “FEV1/FVC” necessary in the title?

Table 4

Align the decimal points of the Positive Likelihood Ratio.

Line 198-199 “although it did not include variables related to skeletal muscle mass”

In the ROC analysis, the JARN model is analyzed with FVC and the Kera model with PEFR. Therefore, both models were analyzed without considering skeletal muscle mass in ROC, and the Kera model’s expression “although it did not include variables related to skeletal muscle mass” may not be appropriate.

Line 210-211

I would like to see a clearer reason for recommending VC compared to FVC.

Author Response

We wish to express our strong appreciation to the reviewers for their insightful comments on our paper. We feel the comments have helped us significantly improve the paper.

Reviewer 2

Comments and Suggestions for Authors

Thank you for inviting me to review this manuscript on “Comparison of Characteristics of Definition Criteria for Respiratory Sarcopenia.” I think the topic is very important for readers in the field of geriatric. However, there are several significant concerns in the study.

Response: Thank you for your suggestions.

Line 24 Is “frailty” correct? In Table 3, female frailty is not significant.

Response: We apologize for this error. The performance for sarcopenia is significant only in FVC of women, which is also a very ineffective result. We have revised the abstract accordingly (line 24).

Line 26 There is no description of what “higher than” is about.

Response: “Sensitivity” was omitted here. We have added it (line 26).

Line 59 Although MIP/MEP is mentioned, MEP is not included in the method of evaluation of respiratory sarcopenia according to reference number 13.

Response: Thank you for your comment. However, JARN recommends the measurement of respiratory muscle strength by maximal mouth pressure to diagnose respiratory sarcopenia (see Fig. 2 in Nagano’s paper).

Line115-117 It would be easier for readers to understand if the specific formula for the cutoff value of FVC is described. Please consider it.

Response: This formula is very complex. LLN is calculated using an Excel sheet according to a previous study.

Table 3 Is “FEV1/FVC” necessary in the title?

Response: Thank you for your suggestion. As is well known, FEV1/FVC is a definite indicator of airway obstruction and is not considered in the current studies.

Table 4 Align the decimal points of the Positive Likelihood Ratio.

Response: Thank you for the insightful comment. As you pointed out, the position of the decimal point was wrong. We have corrected this in Table 4.

Line 198-199 “although it did not include variables related to skeletal muscle mass”

In the ROC analysis, the JARN model is analyzed with FVC and the Kera model with PEFR. Therefore, both models were analyzed without considering skeletal muscle mass in ROC, and the Kera model’s expression “although it did not include variables related to skeletal muscle mass” may not be appropriate.

Response: Certainly, ROC analysis does not use skeletal muscle mass; however, the sensitivity and specificity of respiratory sarcopenia defined by the JARN (using skeletal muscle) and Kera criteria were calculated for sarcopenia, LTCI, and frailty.

Line 210-211 I would like to see a clearer reason for recommending VC compared to FVC.

Response: We have revised this sentence according to your comment (lines 228-234).
